# Cross-Generational Impact of Innate Immune Memory Following Pregnancy Complications

**DOI:** 10.3390/cells11233935

**Published:** 2022-12-06

**Authors:** Nakeisha A. Lodge-Tulloch, Alexa J. Toews, Aline Atallah, Tiziana Cotechini, Sylvie Girard, Charles H. Graham

**Affiliations:** 1Department of Biomedical and Molecular Sciences, Queen’s University, Kingston, ON K7L 3N6, Canada; 2Department of Obstetrics and Gynecology, Department of Immunology, Mayo Clinic, Rochester, MN 55905, USA

**Keywords:** DOHaD, innate immune memory, trained immunity, pregnancy complications, inflammation, cross-generational

## Abstract

Pregnancy complications can have long-term negative effects on the health of the affected mothers and their children. In this review, we highlight the underlying inflammatory etiologies of common pregnancy complications and discuss how aberrant inflammation may lead to the acquisition of innate immune memory. The latter can be described as a functional epigenetic reprogramming of innate immune cells following an initial exposure to an inflammatory stimulus, ultimately resulting in an altered response following re-exposure to a similar inflammatory stimulus. We propose that aberrant maternal inflammation associated with complications of pregnancy increases the cross-generational risk of developing noncommunicable diseases (i.e., pregnancy complications, cardiovascular disease, and metabolic disease) through a process mediated by innate immune memory. Elucidating a role for innate immune memory in the cross-generational health consequences of pregnancy complications may lead to the development of novel strategies aimed at reducing the long-term risk of disease.

## 1. Introduction

Complications of pregnancy, such as miscarriage, preterm birth, pre-eclampsia (PE), and fetal growth restriction (FGR), can have serious immediate and long-term negative consequences to the health of the affected mothers and their children [1,2,3,4,5,6]. These complications are associated with an exaggerated maternal immune response [7,8], and epidemiological data have revealed that mothers and children exposed to aberrant inflammation during pregnancy have an increased risk of developing various inflammation-related conditions, such as cardiovascular and metabolic disease, later in life [6,9].

The innate immune system contributes significantly to the establishment of normal pregnancy [10,11], and maladaptation of the innate immune system has been implicated in the pathophysiology of many severe pregnancy complications [12,13,14,15]. There is also evidence that the innate immune system can acquire memory in response to inflammatory stimuli, which in turn has been postulated to contribute to the development of diseases [16]. For example, trained immunity (TI), a form of innate immune memory, has been linked to the development of various diseases, including autoimmune diseases, Alzheimer’s disease, and atherosclerosis [17,18,19]. In this review, we discuss the potential role of innate immune memory in the cross-generational development of noncommunicable diseases (i.e., pregnancy complications, cardiovascular disease, and metabolic disease) following pregnancy complications.

## 2. Inflammation and the Developmental Origins of Health and Disease Hypothesis

Pregnancy is a process associated with systemic physiological adaptations [20]. Thus, deviations from normal adaptations can predispose mothers and offspring to disease [21,22,23,24]. The developmental origins of health and disease hypothesis, first proposed by David Barker in the early 1990s [25,26], states that environmental insults during critical developmental periods can induce adaptations that allow an individual to survive these challenges. Although necessary to ensure short-term survival, these adaptations may become maladaptive later in life, thereby increasing the risk of developing noncommunicable diseases. Notably, there are various in utero stressors that affect the long-term health of the offspring. Exposure to famine has been associated with an increased risk of future diseases including coronary heart disease (CHD) and metabolic-associated fatty liver disease [27,28]. Furthermore, maternal psychological stress during pregnancy has been linked to the development of neuropsychiatric disorders, such as schizophrenia and psychosis, in the offspring [29,30], an effect that may be mediated by maternal immune activation and inflammation [31].

Various complications of pregnancy are characterized by underlying inflammation that alters the in utero environment and influences the development of future disease in the offspring [8]. Pre-eclampsia, FGR, and preterm birth have also been associated with an increased risk of overt cardiovascular disease (CVD) and CVD risk factors in the offspring [32,33,34]. Furthermore, FGR has been linked to impaired lung function, insulin resistance, and non-alcoholic fatty liver disease in childhood [35,36,37], as well as the development of type 2 diabetes and fatty liver disease in adulthood [38,39]. Some pregnancy complications are also associated with future reproductive complications in the affected offspring. Studies have found that daughters born to pre-eclamptic mothers have an increased risk of developing pre-eclampsia when they become pregnant, and sons born to pre-eclamptic women also have a higher risk of fathering a pregnancy complicated by pre-eclampsia [40,41,42]. Additionally, pre-term female offspring are more likely to experience a pregnancy complication during their reproductive years when compared with female offspring born at term [1].

We recently reported that aberrant maternal inflammation affects not just the offspring directly exposed, but also subsequent generations in the absence of overt inflammatory stimuli [43]. Using a rat model of a pre-eclampsia-like syndrome induced by the administration of low-dose lipopolysaccharide (LPS) during the second half of pregnancy, we demonstrated that pups born to daughters that were exposed to aberrant inflammation in utero were significantly growth-restricted when compared with controls [43]. Uteroplacental units from these pups exhibited reduced GLUT-1 expression and increased numbers of CD68^+^ macrophages [43]. Thus, the consequences of aberrant maternal inflammation in pregnancy may cross generations and impact placental transporter expression, immune cell populations, and fetal growth.

Beyond overt pregnancy complications, other forms of inflammation during pregnancy, such as inflammation associated with obesity, type 2 diabetes, autoimmune disorders, and maternal infection, negatively affect the long-term health of the offspring [44,45,46,47]. Obesity induces a state of chronic, low-grade systemic inflammation through an increased release of proinflammatory cytokines such as tumor necrosis factor alpha (TNF), interleukin (IL)-1β, and leptin [48]. Children of obese mothers are at an increased risk of developing childhood asthma [49] and neurodevelopmental delays associated with a lower IQ [50]. In adulthood, maternal obesity during pregnancy is linked to an increased risk of all-cause death, cancer, CHD, stroke, and diabetes in the offspring [44]. Type 2 diabetes is also characterized by systemic inflammation [51] and has been associated with a variety of long-term health consequences in the offspring, such as an increased risk of childhood asthma [52] and psychiatric disorders in adulthood [46]. Pregestational diabetes, including type 1 and type 2 diabetes, as well as gestational diabetes, were associated with early onset cardiovascular disease in the offspring [53]. Additionally, offspring born to mothers with autoimmune diseases, such as type 1 diabetes and rheumatoid arthritis, had an elevated risk of developing mental disorders including schizophrenia, obsessive compulsive disorder, and mood disorders [45]. Furthermore, inflammation due to infection during pregnancy, such as during chlamydia infection and chorioamnionitis, has been associated with an increased risk of childhood asthma in the offspring [47,54].

## 3. Immunology of Normal Pregnancy

The inflammatory profile of normal pregnancy is characterized by three immunological phases [11]. The first phase is described as a systemic pro-inflammatory or T-helper cell type 1 (Th1) state [11,55]. This phase consists of high levels of circulating maternal inflammatory cytokines including IL-2, IL-8, and IL-17, in addition to the processes associated with establishing pregnancy, such as implantation, trophoblast invasion, and spiral artery remodeling mediated primarily by the invading extravillous trophoblast cells [56,57]. The remodeling of the spiral arteries is critical for the establishment of adequate uteroplacental hemodynamics and, consequently, optimal fetal and placental development [58,59]. The second immunological phase of pregnancy consists of an immune tolerant state characterized by a T-helper cell type 2 (Th2) profile [55], which involves the production of anti-inflammatory cytokines such as IL-4, IL-10, and IL-13 by Th2 cells [60]. Moreover, this phase is defined by a decrease in the circulating levels of pro-inflammatory cytokines, thereby facilitating fetal growth and placental maturation [57]. The final immunological phase is a pro-inflammatory state, which is required for the initiation of labor [11].

Maternal innate immune cells, such as monocytes, decidual macrophages, and uterine natural killer (uNK) cells, are crucial for the establishment and maintenance of pregnancy and contribute to each immunological phase both systemically and locally at the maternal–fetal interface [61,62]. Circulating monocytes can be classified as classical or nonclassical, with the proportion of non-classical monocytes increasing in gestation [61,63]. Additionally, monocytes contribute to the pro-inflammatory environment during the third trimester and facilitate parturition [61]. Macrophages, cells derived from the differentiation of monocytes, also have important roles in pregnancy. These cells comprise 20–30% of the leukocyte population at the maternal–fetal interface and can have a pro-inflammatory (M1) or anti-inflammatory (M2) phenotype throughout pregnancy [64]. Furthermore, a balance of decidual M1 and M2 macrophages is important for establishing and sustaining maternal–fetal tolerance, the remodeling of the spiral arteries, and trophoblast invasion [61,62,64]. Uterine natural killer (uNK) cells localized to the decidua basalis of the maternal–fetal interface also play an important role in pregnancy [65]. These cells comprise 70% of the resident lymphocytes in the decidua during early pregnancy, with numbers decreasing after the first trimester [66,67,68]. In addition to monocytes, uNK cells contribute to the maintenance of maternal–fetal tolerance through immunomodulation at the maternal–fetal interface [69,70,71]. The production of interferon (IFN)γ by uNK cells is also important for the adequate remodeling of the spiral arteries in mice [72]. It is evident, therefore, that the immune system is critical for establishing a successful pregnancy. Consequently, dysregulation of these phases and alterations in the numbers and proportions of circulating and resident cell populations may lead to pregnancy complications.

## 4. Immunology of Pregnancy Complications

As stated earlier, some of the most severe complications of pregnancy are characterized by a deviation from the immunological profile of normal pregnancy [73]. Furthermore, specific complications often have immunological features that distinguish them from other complications.

### 4.1. Miscarriage

Miscarriage is defined as a loss of pregnancy before 20 weeks of gestation and occurs in approximately 10% of all clinically recognized pregnancies [74]. Advanced maternal age, a high body mass index, and a history of pregnancy loss are independent factors contributing to an increased risk of miscarriage [75].

Studies have shown that high levels of inflammatory cytokines systemically and locally at the maternal–fetal interface are associated with the pathogenesis of recurrent miscarriage, defined as three or more consecutive miscarriages [76,77,78,79]. For example, at the maternal–fetal interface, miscarriage was shown to be associated with an increase in the numbers of Th1 cells, which produce pro-inflammatory cytokines such as IFNγ and IL-2, and a decrease in the levels of anti-inflammatory cytokines such as IL-4 and IL-10 [76,80]. Another study revealed a similar shift toward a Th1 immune profile in miscarriages versus normal pregnancies that included an increase in the circulating levels of TNF, IL-6, and IFNγ [79]. Furthermore, the median levels of TNF produced by mitogen-stimulated peripheral mononuclear blood cells (PBMCs) from women who experienced recurrent miscarriage were increased compared to first-trimester PBMCs obtained from women with uncomplicated pregnancies [81]. In murine models, TNF contributes to inflammation-induced fetal loss [82] via mechanisms mediated by uNK cells [83].

### 4.2. Preterm Birth

Preterm birth affects 5–18% of pregnancies and 15 million babies every year [84,85]. It is defined as a birth that occurs before 37 weeks of gestation and is subclassified based on gestational age, with severity ranging from extremely premature to late preterm [84]. As with miscarriage, a higher risk of preterm birth is associated with increasing maternal age [86], and babies born prematurely are at a higher risk of developing chronic conditions such as neurological and respiratory diseases [87,88].

While normal parturition is associated with a pro-inflammatory state [89], preterm birth is associated with the upregulation of pro-inflammatory cytokines, such as IL-1β and TNF, and the downregulation of the anti-inflammatory cytokine IL-10 [14]. Damage-associated molecular patterns (DAMPs) and pathogen-associated molecular patterns (PAMPs) activate and engage pattern recognition receptors (PRRs), such as toll-like receptors (TLRs), on uNK cells [90,91,92]. This activation has been linked to the increased inflammation that characterizes some cases of preterm birth [90,91,93]. Furthermore, preterm birth has been associated with the upregulation of over 100 differentially expressed genes when compared to term pregnancies [94]. These genes were enriched for inflammatory pathways and contribute to immune regulation [94].

### 4.3. Pre-Eclampsia

Pre-eclampsia is a major cause of maternal morbidity and mortality, affecting approximately 5% of pregnancies [95,96,97]. Pre-eclampsia is characterized by de novo maternal hypertension and end-organ damage occurring after 20 weeks of gestation [96,98]. Additionally, one third of pre-eclampsia cases are also complicated by FGR [99]. Several factors are associated with an increased risk of pre-eclampsia; these include maternal age, multifetal pregnancy, and paternal specificity [97,100]. Pre-eclampsia is a heterogeneous syndrome, and several placental subtypes have been implicated in its pathogenesis, including a ‘canonical’ subtype and an ‘immunological’ subtype [101]. The canonical subtype is characterized by maternal vascular malperfusion, resulting in placental tissue damage due to a hypoxic environment, and the release of placental products (e.g., DAMPs) into the maternal circulation [101]. The immunological subtype has been described as an infiltration of various immune cells, including monocytes and neutrophils, into the intervillous space [102]. This subtype is further characterized by histological evidence of chronic inflammation characterized by an increase in CD8^+^ T-cells, CD68^+^ macrophages/monocytes, and myeloperoxidase-expressing neutrophils [102]. Additionally, there is evidence of the upregulation of genes associated with homeostasis, immune response, inflammatory response, and cytokine activity [101].

Severe maternal inflammation resulting from the abnormal activation of immune cells may contribute to the pathogenesis of pre-eclampsia [103]. One proposed hypothesis is that increased numbers of cytolytic NK (cNK) cells contribute to the development of pre-eclampsia [103,104]. These cells differ from regulatory NK cells based on their CD56 and CD16 expression, whereby cNK have a CD56^dim^CD16^+^ phenotype and regulatory NK cells are characterized by CD56^bright^CD16^−^ [104]. Furthermore, a Th1/Th2 cell ratio imbalance that favors Th1 cells, as well as increased levels of pro-inflammatory cytokines such as TNF, may lead to the pro-inflammatory state that characterizes pre-eclampsia [105,106].

### 4.4. Fetal Growth Restriction (FGR)

Fetal growth restriction is defined as the failure of a fetus to reach its full growth potential and affects 10% of pregnancies [107]. FGR is associated with many chronic health conditions, including diminished growth, neurodevelopmental disorders, and persistent immunological impairment [108,109].

As with other common complications of pregnancy, there is a well-described link between FGR and aberrant maternal inflammation [7,110,111]. Using a murine model of inflammation-induced FGR, Cadaret et al. reported in utero skeletal muscle changes that ultimately resulted in FGR, thus demonstrating an association between aberrant maternal inflammation and the pathophysiology of FGR [110]. Moreover, compared with normal human pregnancy, pregnancies complicated by FGR with placental insufficiency exhibit increased maternal levels of the proinflammatory cytokines IL-8 [112] and TNF [111]. Using an inflammation-induced rat model of PE-like disease, we demonstrated a causal link between TNF and impaired uteroplacental perfusion associated with FGR [7]. Additionally, prenatal uric acid exposure resulted in lasting FGR and long-term neurodevelopment defects through placental inflammation [113].

## 5. Placental Stress and Release of Damage-Associated Molecular Patterns

While complications of pregnancy can exhibit unique immunological profiles, a pro-inflammatory environment characterized by increased levels of TNF is a shared feature of these conditions [14,79,81,105,111]. Furthermore, these complications are associated with altered uteroplacental hemodynamics, which are often attributed to abnormal placentation [7,73,82,114,115,116,117]. In addition, the maternal pro-inflammatory milieu typical of pregnancy complications may directly contribute to altered placental perfusion. For instance, TNF is a well-known vasoconstrictor, and TNF-activated signal transduction pathways have been shown to contribute to vascular dysfunction [118]. Together, these findings point to a strong causal association between aberrant inflammation and altered uteroplacental hemodynamics characteristic of severe complications of pregnancy.

Altered uteroplacental hemodynamics can result in placental hypoxia [119], oxidative stress [7,120,121], and physical damage to the maternal–fetal interface due to increased uteroplacental perfusion pressure [120]. These insults to placental tissues cause the release of placental-derived DAMPs, including heat shock proteins, uric acid, ATP, and high-mobility group box-1 (HMGB1) proteins, into the maternal circulation [122]. The activation of PRRs by DAMPs results in the initiation of signaling pathways such as the myeloid differentiation factor 88 (MyD88)-dependent and -independent pathways, ultimately leading to the production of inflammatory cytokines [123]. Notably, TLR-4 plays a role in the pathophysiology of preterm birth, pre-eclampsia, and FGR [124,125,126]. The binding of agonists to TLR-4 receptors on uNK cells results in the association of the TLR-4 receptor with MyD88 or the adaptor protein Toll/IL-1 receptor domain-containing adaptor inducing IFN-β (TRIF), ultimately activating the NFκB pathway [123]. This eventually stimulates the production of pro-inflammatory cytokines, such as IFNγ, by uNK cells [127]. In pregnancy complications such as pre-eclampsia and preterm birth, TLRs can be overexpressed, leading to excessive inflammation [125,128].

We propose that during a pregnancy complication, the excessive release of placental DAMPs due to uteroplacental malperfusion leads to the amplification of the initial inflammation that triggered the pathogenesis of the complication. We also propose that the enhanced release of placental DAMPs in the maternal circulation is critical for long-term epigenetic reprogramming and memory acquisition in bone marrow innate immune cell precursors, leading to an increased risk of disease later in life.

## 6. Adaptations of the Innate Immune System

The innate immune system adapts to insults including infections, inflammation, and injury [129]. Adaptive programs include differentiation, priming, and the acquisition of trained immunity (TI) and tolerance, and depend on the duration, type, and magnitude of the insult [129]. Differentiation is associated with a change in the function and morphology of cells from immature to mature phenotypes [129]. Priming includes functional changes in phenotype leading to enhanced immunological activity [129]. Once the stimulus is removed, this activity remains constant and does not return to basal levels. Priming contrasts with innate memory, which is characterized by either an enhanced (trained immunity) or a dampened (tolerance) production of pro-inflammatory cytokines upon a secondary exposure to a similar but not necessarily identical DAMP or PAMP (Figure 1) [129].

Trained immunity and tolerance occur due to epigenetic changes, e.g., histone modifications, leading to the enhanced or decreased transcriptional activation of pro-inflammatory genes, respectively, upon subsequent exposure to a DAMP or a PAMP [129]. For example, studies have demonstrated that when innate immune cells are trained and active, the trimethylation of the fourth lysine in histone H3 (H3K4me3) is increased throughout the genome, indicating an open chromatin configuration [130]. When the cell returns to basal activation levels, the monomethylation of lysine 4 in histone 3 (H3K4me1) persists and allows for a rapid and enhanced response upon restimulation [130]. These TI-associated epigenetic modifications result in the enhanced production of proinflammatory cytokines (e.g., TNF and IL-1β) [130]. The epigenetic changes that occur in innate immune cells depend on the quantity and type of PAMPs and DAMPs that interact with their respective PRRs [19]. For example, when LPS binds to TLR-4, tolerance to endotoxin can be induced [19]. Indeed, Wen et al. reported that peritonitis-induced sepsis led to long-term immunosuppression in dendritic cells as a result of epigenetic changes [131]. However, tolerance has also been demonstrated to be reversible. Ifrim et al. demonstrated that the exposure of monocytes to high concentrations of TLR ligands leads to tolerance, which can be switched to TI upon exposure to low concentrations of TLR ligands [132]. Moreover, LPS-induced tolerance in monocytes can be reversed following exposure to β-glucan (a component of the cell wall of *Candida albicans*), a PAMP associated with TI that binds primarily to the PRR dectin-1 [133].

### 6.1. Trained Immunity: Friend or Foe?

TI was first described following the observation of the non-specific beneficial effects of vaccines that could not be solely attributed to the adaptive immunological memory [134]. For example, Bacillus Calmette–Guérin (BCG) inoculation, a live attenuated mycobacterium used as a vaccine for tuberculosis, resulted in a decreased rate of mortality—mainly by decreasing the rate of respiratory infections and sepsis—within days of vaccination, a time frame too short for adaptive immunity to develop [135]. Other vaccines, such as yellow fever, vaccinia, and measles vaccines, were also shown to exhibit non-specific beneficial effects by protecting against several diseases/infections as well as reducing overall morbidity and mortality rates [136]. Moreover, studies have shown that TI following vaccination with BCG protects against yellow fever [137] and malaria [138]. Trained immunity inducers include various types of PAMPs and DAMPs, such as BCG, which is one of the most studied PAMPs in the trained immunity field, along with β-glucan [139,140]. Innate lymphoid cells such as NK cells as well as myeloid phagocytes such as monocytes, macrophages, and neutrophils have all been shown to acquire TI [141]. Although trained immune cells in circulation have a relatively short half-life (e.g., 2–5 days for monocytes), this form of memory has been shown to persists for several months, indicating that it must be acquired at the level of hematopoietic stem and progenitor cells [19].

In most cases, TI is considered to be a protective mechanism against infections and diseases such as cancer. However, there is evidence that TI may also contribute to disease. For instance, a study by Bekkering et al. revealed that, compared with monocytes from healthy individuals, monocytes from patients with symptomatic atherosclerosis release higher levels of proinflammatory cytokines upon in vitro exposure to a secondary inflammatory stimulus, indicating evidence of TI [17]. Monocytes from these patients also exhibited the increased expression of glycolytic enzymes, which is associated with epigenetic reprogramming at the histone methylation level, hence indicating higher levels of TI compared with monocytes from asymptomatic individuals [17]. Moreover, hyperglycemia induces TI in macrophages and subsequently promotes atherosclerosis [142]. Similarly, TI may be associated with various cardiovascular disorders [142]. Recent evidence also implicates TI in allograft rejection due to the increased production of proinflammatory cytokines by trained macrophages, leading to the activation of the adaptive immune response [143]. Additionally, other conditions associated with TI acquisition include hyper-IgD syndrome; autoinflammatory disorders such as familial fever syndromes, gout, and inflammatory bowel disease; type 2 diabetes; and systemic lupus erythematosus [19]. Similarly, mouse models of Alzheimer’s disease revealed evidence of TI and tolerance in microglia [18].

### 6.2. Cross-Generational Inheritance of Trained Immunity

Epigenetic signatures can be inherited, which raises the possibility that TI can also be transmitted cross-generationally and therefore play an important role in the transmission of disease predisposition from mother to child. One possible mechanism for the transmission of TI involves the epigenetic reprogramming of the gametes following the exposure of one or both parents to inflammation. A study by Katzmarski et al. revealed evidence of TI transmission and heterologous resistance to infections across generations [144]. Male mice that survived a sublethal infection with *Candida* or a zymosan challenge, and developed TI as a result, were able to transmit resistance to infection by unrelated bacteria (*E. coli* or *L. monocytogenes*) to the F1 progeny [144]. This resistance was mediated mainly through the enhanced activation of myeloid cells. While that study provided evidence of the cross-generational transmission of protective TI, one may also infer that pathological TI can be inherited in a similar way. However, the findings from the study by Katzmarski et al. were not reproduced by Kaufmann et al., who described a lack of cross-generational inheritance of immune resistance to infections using a similar mouse model [145].

Another potential mechanism is that the uteroplacental unit is key to the transmission of TI from mother to offspring. Altered uteroplacental hemodynamics resulting from inflammation could lead to the release of placental DAMPs not just in the maternal blood, but also in the fetal circulation. The presence of DAMPs in the fetal circulation could then lead to epigenetic reprogramming and the acquisition of memory in fetal bone marrow myeloid precursors. In support of this hypothesis are findings from our laboratory revealing the increased presence of inflammatory macrophages in the uteroplacental units from pregnant rat offspring previously exposed to LPS during in utero development [43]. Moreover, pups from rats exposed to LPS in utero were growth-restricted, indicating that the risk of disease is also transmitted across generations, likely via inflammatory memory [73].

## 7. Potential Mechanisms Linking Acquisition of TI after Exposure to Aberrant Maternal Inflammation to Increased Risk of Disease

### 7.1. Predisposition to Disease Due to Underlying Conditions

There is a general consensus that pregnancy is a physiological ‘stress test’ that unmasks underlying conditions or risk factors that develop into overt disease later in life [146]. This conclusion is based on the observation that pregnancy complications often share etiologies with chronic inflammation-associated diseases later in life [146]. For example, as pregnancy is characterized by a pro-thrombotic state, hereditary thrombophilia is often unmasked by venous thrombosis in pregnancy [146]. Pregnancy can also trigger or exacerbate symptoms of undiagnosed lymphangioleiomyomatosis, a rare cystic lung disease that affects premenopausal women [147,148]. Thus, pregnancy can be viewed as a stress test to identify risks of future disease and is therefore a critical period to implement primary preventative care and enhanced follow-up measures for women at high risk for certain diseases, such as cardiovascular disease [149]. However, the cross-generational impacts of these underlying conditions and their relation to future health in the offspring remain unclear. Contrastingly, one can hypothesize that instead of unmasking a disease, pregnancy complications act as secondary stimuli that exacerbate the TI acquired due to an underlying condition. Consequently, the development of disease later in life may be mediated by TI.

It is also important to consider the impact of genetic factors and lifestyle choices on the development of pregnancy complications and the risk of subsequent disease. For example, the polygenic risk score for high blood pressure is associated with an increased risk of pre-eclampsia, as well as the development of hypertension [150]. These genetic risk factors may also be transmitted to the offspring, thereby predisposing them to future disease. Diet has also been associated with complications in pregnancy and offspring health. For example, the high maternal intake of saturated fats is associated with pre-eclampsia, FGR, and preterm birth [151]. Dietary saturated fats are also linked to cardiovascular disease [152]. Parental food habits are the most significant determinants of children’s eating behavior [153]; therefore, parents may model the consumption of diets linked to future disease and influence their children’s diets. Because childhood dietary patterns can persist into adulthood [154], the social inheritance of diets associated with disease may further contribute to the development of noncommunicable diseases in adulthood.

### 7.2. TI as a Mediator of Cross-Generational Disease

Given the evidence that TI and the effects of pregnancy complications persist across generations, it is possible that TI is causally linked to the increased risk of offspring disease following exposure to aberrant inflammation (Figure 2). However, much remains to be elucidated in this field, and whether an increased risk of pregnancy complications and subsequent disease is mechanistically linked to the acquisition of TI has yet to be determined.

Gamliel et al. investigated the acquisition of TI in uNK cells and their function in subsequent pregnancies [155]. The investigators reported that trained uNK cells obtain physiological memory during normal pregnancy [155]. These cells, which the authors termed pregnancy-trained decidual NK (PTdNK) cells, were characterized by the high expression of the NKG2C and LILRB1 receptors [155]. These PTdNK cells persist in subsequent pregnancies and exhibit the increased secretion of the proinflammatory cytokines IFNγ and vascular endothelial growth factor (VEGF), which are known to promote placental vascularization and, consequently, pregnancy success [155].

In another study, Dang et al. explored the impacts of pathological memory on pregnancy by investigating the effect of BCG as a PAMP to induce TI [156]. These investigators established a mouse model that involved the administration of BCG multiple times prior to pregnancy. They found that establishing TI before pregnancy led to FGR, although they did not observe abnormal placental histology or gene expression at the end of gestation when comparing ‘trained’ and ‘untrained’ mice [156]. However, that study did not provide data on TI-associated cytokine production by circulating immune cells.

## 8. Conclusions

Complications during pregnancy and other inflammatory conditions that predispose offspring to future disease are associated with a deviation from the normal immunological signature of pregnancy. Therefore, we propose that immune maladaptation during pregnancy results in the acquisition of innate immune memory that can persist cross-generationally, affecting the long-term health of the offspring.

Consequently, TI may serve as an indicator of increased risk for future noncommunicable diseases in the offspring. The extent of TI could be determined by assessing epigenetic modifications and/or cytokine release in myeloid or lymphoid innate immune cells. Therefore, the acquisition of TI could be monitored as a way of identifying children at high risk of developing disease later in life. Guided therapies could be developed to eliminate and/or mitigate the acquisition of TI and, by association, the increased risk of disease in subsequent generations.

## Figures and Tables

**Figure 1 cells-11-03935-f001:**
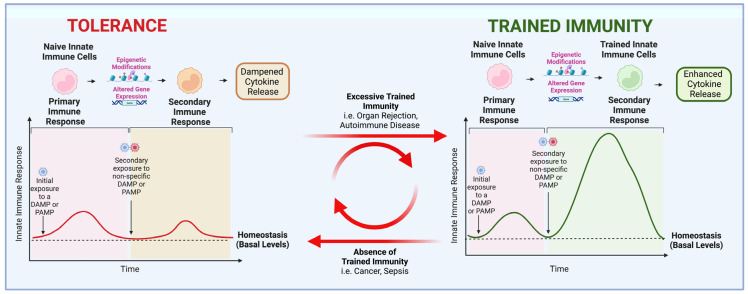
Trained immunity versus tolerance. Trained immunity and tolerance are examples of adaptations of the innate immune system. In response to initial exposure to damage-associated molecular patterns (DAMPs) or pathogen-associated molecular patterns (PAMPs), naïve innate immune cells respond with a release of pro-inflammatory cytokines. When the stimulus is removed, cells return to basal-level activity, but epigenetic signatures persist. Upon secondary exposure to a non-specific DAMP or PAMP, cells that have acquired epigenetic signatures associated with tolerance respond with decreased cytokine release, whereas those with trained immunity signatures respond with enhanced cytokine release. Several disorders have been linked to excessive trained immunity acquisition, including organ transplant rejection and autoimmune diseases, while others have been linked to the absence of trained immunity, such as cancer and sepsis.

**Figure 2 cells-11-03935-f002:**
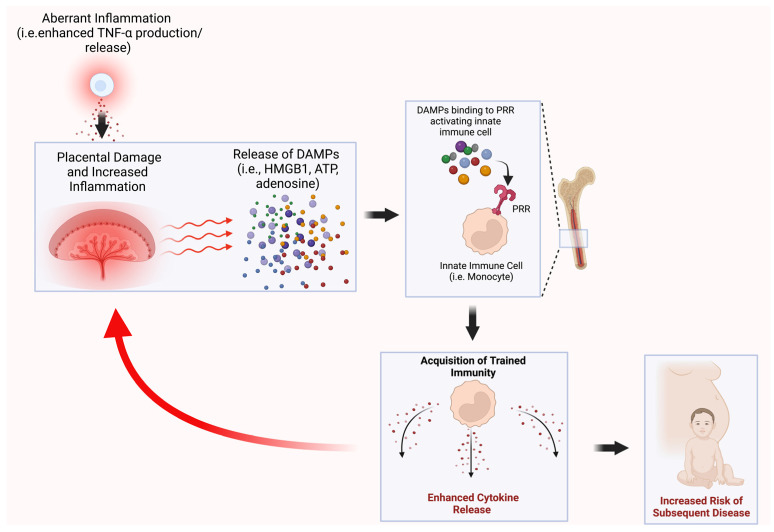
Potential mechanism linking the acquisition of trained immunity after exposure to aberrant maternal inflammation to an increased risk of disease. We hypothesize that the aberrant maternal inflammation associated with pregnancy complications results in the acquisition of trained immunity (TI) in mothers and their offspring, and that this TI mediates the increased risk of adverse health outcomes in subsequent generations. DAMPs—damage-associated molecular patterns; HMGB1—high-mobility group box 1; ATP—adenosine triphosphate; PRR—pattern recognition receptor.

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
