# Peer review of "Cross-Generational Impact of Innate Immune Memory Following Pregnancy Complications"

_cells, 2022, doi:10.3390/cells11233935_

Round 1

Reviewer 1 Report

Title: This is succinct

Abstract: the abstract clearly summarizes the article topic of common pregnancy complications influencing the risk of the next generation developing NCDs mediated by innate immune memory.

Body of the manuscript 

1. The Introduction section is clearly written. Minor suggestion: since much of the intro focuses on immune system/inflammation and future cardiometabolic disease, possibly the mention of Alzheimer’s should be removed to maintain focus.

2. Inflammation and the Developmental Origins of Health and Disease Hypothesis, 2nd paragraph: Assuming FGR is fetal growth restriction, but this should be spelled out at first use.

Overall, well written summary of a highly relevant topic related to pregnancy outcomes, as well as maternal and child future disease risk. 

Author Response

Title: This is succinct

Abstract: the abstract clearly summarizes the article topic of common pregnancy complications influencing the risk of the next generation developing NCDs mediated by innate immune memory.

Body of the manuscript

  1. The Introduction section is clearly written. Minor suggestion: since much of the intro focuses on immune system/inflammation and future cardiometabolic disease, possibly the mention of Alzheimer’s should be removed to maintain focus.

We thank this reviewer for the constructive comments. The role of TI in the development of Alzheimer’s disease is also discussed in Section 6.1 (lines 332-334). Therefore, we believe we need to keep it also in the Introduction.

  1. Inflammation and the Developmental Origins of Health and Disease Hypothesis, 2nd paragraph: Assuming FGR is fetal growth restriction, but this should be spelled out at first use.

FGR was defined earlier in line 27 of the manuscript.

Overall, well written summary of a highly relevant topic related to pregnancy outcomes, as well as maternal and child future disease risk.

Reviewer 2 Report

In the paper entitled “Cross-generational impact of innate immune memory following pregnancy complicationsthe authors review and discuss the potential role of innate immune memory in the cross-generational development of noncommunicable diseases following pregnancy complications. The review is interesting. I would recommend the authors to address the following points when revising the manuscript.

1. In this manuscript, the authors focus on a number of pregnancy-related disorders. Are the severity of these diseases or the different gestational periods related to the immune memory and cross-generational impact mentioned by the authors?

2. I would appreciate the author providing more examples of innate immune memory related to pregnancy-related diseases and cross-generational impact.

3.Consequently, TI may serve as an indicator of the risk of future noncommunicable diseases in the offspring…… at high risk of developing disease later in life. Is there any specific indicator for TI?

Author Response

Comments and Suggestions for Authors

In the paper entitled “Cross-generational impact of innate immune memory following pregnancy complications ”,the authors review and discuss the potential role of innate immune memory in the cross-generational development of noncommunicable diseases following pregnancy complications. The review is interesting. I would recommend the authors to address the following points when revising the manuscript.

  1. In this manuscript, the authors focus on a number of pregnancy-related disorders. Are the severity of these diseases or the different gestational periods related to the immune memory and cross-generational impact mentioned by the authors?

This is an interesting comment. This field, as it relates to pregnancy complications, is largely unexplored. It is not known whether the severity of pregnancy complications impact the extent of innate immune memory acquired. It is also unknown whether the extent of innate immune memory relates to the severity of subsequent disease.

  1. I would appreciate the author providing more examples of innate immune memory related to pregnancy-related diseases and cross-generational impact.

 Unfortunately (or fortunately, depending on one’s perspective), there are no known examples on how innate immune memory is related to pregnancy complications and its cross-generational impact. We commented on this in the manuscript (lines 397-400). This lack of information on this topic is what makes this review timely and impactful.

3.“Consequently, TI may serve as an indicator of the risk of future noncommunicable diseases in the offspring…… at high risk of developing disease later in life”. Is there any specific indicator for TI?

In lines 272-274, we indicate that TI-associated epigenetic modifications result in enhanced production of proinflammatory cytokines. Consequently, one would expect that TI acquisition could be measured in the offspring via assessment of epigenetic changes and/or cytokine release by innate immune cells. We have added a sentence in the Conclusion section to clarify this.